# Evolution as a Guide to Designing *xeno* Amino Acid Alphabets

**DOI:** 10.3390/ijms22062787

**Published:** 2021-03-10

**Authors:** Christopher Mayer-Bacon, Neyiasuo Agboha, Mickey Muscalli, Stephen Freeland

**Affiliations:** 1Department of Biological Sciences, University of Maryland, Baltimore County, Baltimore, MD 21250, USA; cmayerb1@umbc.edu (C.M.-B.); nagboha1@umbc.edu (N.A.); 2Individualized Study Program, University of Maryland, Baltimore County, Baltimore, MD 21250, USA; mmuscal1@umbc.edu

**Keywords:** amino acid, ncAA, protein structure prediction, optimality, chemistry space

## Abstract

Here, we summarize a line of remarkably simple, theoretical research to better understand the chemical logic by which life’s standard alphabet of 20 genetically encoded amino acids evolved. The connection to the theme of this Special Issue, “Protein Structure Analysis and Prediction with Statistical Scoring Functions”, emerges from the ways in which current bioinformatics currently lacks empirical science when it comes to xenoproteins composed largely or entirely of amino acids from beyond the standard genetic code. Our intent is to present new perspectives on existing data from two different frontiers in order to suggest fresh ways in which their findings complement one another. These frontiers are origins/astrobiology research into the emergence of the standard amino acid alphabet, and empirical xenoprotein synthesis.

## 1. Introduction

For protein structure prediction involving non-canonical amino acids (ncAA’s [1]), the most recent, significant advance of which we are aware used a highly sophisticated combination of force field libraries and molecular dynamics simulations to predict structures for 551 peptides [2]. Building from an excellent introduction to the state of the field, the authors demonstrated significant improvement over previous work by predicting an impressive “*Deviation between the actual and predicted structures of peptides in the range of 3.81–4.05 Å*”. As one might expect, “*performance [of the algorithm] decreased with the increase in length of the peptide [particularly] when the length of the peptide is >20 residues*” (ibid, page 10 and Table 5). To compare this performance with progress in structure prediction for “natural” proteins, a November 2020 press release [3] announced that a computer program (AlphaFold) given only primary protein sequence data as input could more or less match experimental precision (~1 Å RMSD) in determining the three-dimensional configuration into which this sequence would fold [3]. In this sense, AlphaFold could be said to be ~4 times more accurate than Sing et al.’s algorithm. Subject to further details provided in the forthcoming CASP14 issue of *Proteins*, it appears that AlphaFold can fulfill this potential not just for short peptides but for proteins sequences hundreds of amino acids in length, and across the universe of protein folds. We therefore find no disagreement with the proposition that Alpha Fold’s “*stunning advance … will fundamentally change biological research*” (V. Ramakrishnan, [3]). Everything about this comparison re-affirms the trends and predictions of an authoritative review published 15 years ago regarding the preceding decade of protein structure prediction: “*current major challenges are refining comparative [homology] models [in their approach to] experimental accuracy*” whereas for “*template-free modeling [the need is to] produce more accurate models [in order to] handle parts of comparative models not available from a template*” [4].

When it comes to proteins comprising ncAA’s, however, AlphaFold joins other template-based approaches in offering less help simply because it renders predictions by learning from a subset of the proteins produced by biological evolution. Specifically, each point in this reference library is an amino acid sequence for which careful experimental investigations have revealed a corresponding three-dimensional structure. The challenge is therefore one of applicability domain [5]: The compounds used to train a model define the physicochemical and biological space within which that model’s predictions are most reliable. Machine learning trained on a given library of sequence/structure relationships cannot safely extend predictions to sequences far removed from anything in that library.

The limitations of homology-based template modelling are already real and significant for those who navigate the “*dark matter*” of protein fold space [6], including the theoretical universe of “*never born proteins*” [7]. However, these exciting, overlapping frontiers merit separate, careful discussion in light of the growing awareness of *de novo* proteins by which evolution seems to find novel structures and functions [8]. For present purposes it matters only that the lack of homologous templates applies clearly to proteins comprising ncAA’s. By definition, ncAA’s are amino acids never encountered by biological evolution and for which the very concept of homology is therefore undefined. For the case of relatively few ncAA’s within an otherwise natural protein sequence, wherever template-based approaches use sequence homology, they might reasonably hope to avoid the issue by treating ncAA’s as a “blank” or null character, analogous to the way in which gaps are handled by multiple sequence alignment for homologous proteins of different lengths (e.g., [9]). As the proportion of ncAA’s increases, however, the problem transcends any such fix because the prediction algorithm is left with less legitimate information to work with, and more unknowns which likely influence the fold but are being ignored.

Of course, evolutionary homology is useful to protein fold prediction because it provides an excellent proxy for the physicochemical basis of what Anfinsen discovered. Some machine learning models bypass the need for homology by working directly with this physicochemical basis of protein sequence/structure relationships. Whether representing amino acids directly as a blend of quantitative properties (e.g., [10]) or simplifying the standard amino acid alphabet into abstractions about amino acid similarity [11,12,13], this approach creates a *lingua franca* for all amino acids which offers more promise for handling ncAA’s. However, even here ncAA’s bring new questions about what constitutes the relevant physicochemistry and a subtler extension of the challenge of applicability domain.

In abstract terms, the problem of applicability domain is that in order to extrapolate rules defining interactions between a set of objects safely onto predictions about a superset of those objects, one must assume that the additional members of the superset contain no new types of interaction. Reasons to be cautious about assuming that ncAA’s will bring no new physicochemistry of protein folding are easy to anticipate. Consider, for example, a thought experiment about protein sequence/structure relationships arising from a genetic code which lacked cysteine. Nothing like disulfide bridges would exist to inform us (or a machine learning algorithm) of their existence. It is not clear that a machine learning algorithm would learn to predict this possibility for thermodynamically favorable covalent bond formation between two sulfur atoms from the physicochemical rules learned by studying proteins comprising only the other 19 amino acids. Since disulfide bridges both enabled Anfinsen’s foundational discovery and remain areas of active research when it comes to their role in protein folding [14], it seems pertinent to ask how confident can we be that no further phenomena exist within an indefinitely diverse set of ncAA’s to modify our understanding of sequence/structure relationships? Far less extreme than new covalent bonds, we already know that “*side-chain and backbone interactions [within ‘natural’ protein sequences] may provide the energetic compensation necessary for populating [hitherto unrecognized] region of φ–ψ space*” [15]. Given that empiricists already and routinely incorporate into ribosomal peptide synthesis not only new functional groups, but also new atom types, it would be a bold assumption that proteins as we know them can teach us (or a machine learning algorithm) all that we need to know about physicochemical properties relevant to xenoprotein folds. A closer look at empirical success incorporating ncAA’s demonstrates why the challenge of developing statistical scoring functions for an indefinitely diverse set of ncAA’s is both timely and important.

## 2. Hundreds of ncAAs Have Already Been Incorporated into Proteins

The importance of structure prediction for proteins comprising ncAA’s lies in the rate at which our empirical colleagues are producing them. To date, at least 246 different amino acids from beyond the standard alphabet have been added experimentally into various organisms’ genetic codes (see [16,17,18], Figure 1). The chemical structures involved overlap somewhat with the modified residues considered by Singh et al. [2], but these ncAA’s differ in having been involved with the molecular machinery of gene translation.

Comprehensive information about this fast-growing collection of amino acids is surprisingly sparse. Figure 1 and Figure 2, for example, show data that are unavailable elsewhere as far as we know in that they collate three major reviews of the topic and curate/standardize the chemical structures involved so as to facilitate comparison (see Appendix A). Collectively, these review articles reveal that new ncAA’s are being introduced to proteins at a rate of 10–60 new structures per year since 2006 (Figure 1A). Among much else, this indicates that any statements made now about ncAA structures and biochemistry will have to be reassessed regularly for the foreseeable future. Indeed, 246 residues is likely an underestimate of current progress given the relatively low overlap of ncAA structures reported in common wherever two or more reviews cover the same year (Figure 1B), and the existence of numerous further reviews on a similar topic.

It is true that ncAA’s have thus far usually been added singly or at a few key positions within a “natural” protein sequence, usually for a specialized purpose such as to facilitate detection of the resulting protein by appropriate instrumentation (Figure 2). Different analogues of aromatic amino acids, for example, are a popular choice for introduction of photoactivatable and fluorescent moieties in natural proteins. This description of ncAA usage summarizes, however, only the present situation looking backwards. Polymerizing ncAA’s into a true xenoprotein, one constructed significantly or entirely with amino acids from beyond the standard alphabet is becoming a part of the present looking forwards. In 2019, for example, Feldman et al. reported working with an experimental system in which: “*… a wide variety of unnatural ribonucleotides can be efficiently transcribed into RNA and then … mediate the synthesis of proteins with ncAAs…. The SSO is now, for the first time, able to efficiently produce proteins containing multiple, proximal ncAAs*” [19]. The SSO here refers to a Semi Synthetic Organism—a bacterium that has been designed successfully to incorporate synthetic nucleotides into its genetic material, transcribing these additional genetic letters into new codons which can translate into sequential strings of ncAA’s.

It is worth pausing here to note just how far ahead of protein biochemistry is nucleotide biochemistry when it comes to engineering the parameters with which evolution has worked for most of life’s history. Whereas ncAA’s have started to make their way into natural proteins, already two very different entire genetic alphabets have been developed to the point of in vivo functioning, including replication (Figure 3). The Benner group’s *Hachimoji* alphabet [20] emulates Watson-Crick base pairing using the four atom types known to “natural” nucleotides (N, O, H and C). Both the nucleotide structures and their base-pairing systems look at once familiar and alien to natural genetics (Figure 3A). A second alphabet, developed separately by Romesberg and colleagues, is unquestionably alien in that complementary base “*pairing is mediated not by hydrogen bonding but by hydrophobic and packing forces*” [21]. Of particular note, the Romesberg synthetic nucleobases contain sulfur (Figure 3B), an atom type unknown to natural genetics as Hershey and Chase exploited to earn another of the five Nobel prizes [22] which collectively define the central dogma of molecular biology. The Romesberg alphabet illustrates that synthetic biology may extend chemical structures not just beyond the details of biology, but beyond the fundamental rules as we know them. This idea resonates with the thought experiment of cysteine and disulfide bridges above (what new rules of protein folding might lurk within radically alternative chemical structures?) and gives cause to extend thinking from xenoproteins to xenoalphabets, entire alphabets comprising ncAA’s.

## 3. From Xenoproteins to Xenoalphabets

The challenge for template free models often lies less in the fundamental physicochemical principles involved than the number of possible conformations for which this physicochemistry must be computed. We have learned much since Anfinsen and colleagues performed their Nobel prize-winning insight [23] that “*at least for a small globular protein in its standard physiological environment, the native structure is determined only by the protein’s amino acid sequence*” [24].

We know, for example, that protein folds tend to minimize free energy, often through “collapse” (e.g., [25]) by which hydrophobic amino acid sidechains find one another to form a densely packed hydrophobic core using size/shape mediated packing forces similar to those responsible for base pairing in the Romesberg synthetic nucleotide alphabet (Figure 3B). Charge-charge interactions further stabilize the structure, as do disulfide bridges and a host of further details that can vary in type and significance for different proteins and their cellular contexts and continue to occupy expert research (e.g., [26]). However, what is remarkable is that contemporary computing is often capable of calculating free energy for a given, theoretical protein conformation with a fair degree of accuracy (e.g., see [27]). For example, in 2014 Vreven et al. demonstrated the oversimplification of any simple statement about template-based models outperforming template free alternatives [28]. They compared the performance of two template-based approaches (threading and structure alignment) with a template-free alternative (docking) for the interesting case of protein-protein complexes, which accentuate the difficulties of empirical structure determination. Results showed that template-based methods perform similarly to the template free (docking) alternative when each method is restricted to make a single, best prediction. Template-free docking outperformed template-based methods for “*complexes that involved conformational changes upon binding*”. These findings provide reason for optimism regarding the current situation of a few ncAA’s incorporated into a native protein structure if we think of these introductions as perturbations away from a clear but increasingly misleading template. However, perhaps most interestingly, Vreven et al. conclude, “*(correct) predictions were generally not shared by the various approaches, suggesting that integrating their results could be the superior strategy*”.

This theme of complementary approaches turns our attention from ncAA’s to amino acid alphabets if we return to the more general problem of template free prediction: the number of possible protein conformations that require comparison in order to ascertain which exhibit promising energy minima? In the absence of further information, this number scales exponentially with the length of the protein. As Levinthal [29] is credited with first pointing out, a sequence of 100 amino acids contains 99 peptide bonds, and therefore 198 different φ and ψ bond angles in the peptide backbone. His thought experiment granted only three possible values for each of these angles to define 3^198^ different, possible conformations (including any possible folding redundancy). Ramachandran plots indicate that 3 possible states for each angle is an oversimplification, and it seems likely that here if nowhere else ncAA’s will add further degrees of freedom, given that plausible φ–ψ angles depend on sidechain composition [15]. This accentuates Levinthal’s point: assuming very rapid sampling of each conformation (~1 picosecond), exploring all 3^198^ possible conformations in Levinthal’s example would take longer than the current age of the universe. Given this immense time scale for a simple polypeptide, templates constrain a functional infinity (≫10^198^) of possible configurations to a local neighborhood where careful searching is likely to find relevant free energy minima on a reasonable time scale.

Rational design of individual proteins and subsequent empirical structural analysis combine to provide a direct strategy with which to progress protein structure prediction with statistical scoring functions for ncAA’s (e.g., [30]). Over time this careful approach will not only improve physicochemical calculations to cover new types of atomic interactions but also build a library of templates. Constructing combinatorial protein libraries through biochemical engineering (“total chemical synthesis”) complements this time-consuming, and resource-intensive strategy in these early days of xenoprotein exploration by producing synthetic libraries for specific binding affinities (such as those associated with a given function). Combinatorial libraries provide an objective look at sequence/structure relationships without inheriting biases from other aspects of biology. Gates et al., for example [31], describe how the approach enables identification of small (~30 aa) functional protein variants comprising a virtually unlimited variety of noncanonical amino acids. Direct screening of a synthetic protein library by these methods resulted in the *de novo* discovery of binders to a ~150-kDa protein target, “*a task that would be difficult or impossible by other means*”. This important work leads naturally to the question of which ncAA’s to explore from an indefinitely large chemical space of possibilities? Appropriate choices could usefully inform the relationship between folds and amino acid structures which address a specific structural or catalytic challenge, and their potential to work well within current physicochemical prediction models.

To guide a choice of ncAA alphabets, we suggest here a third layer of theory- rapid, cheap and useful only in the context of the other possibilities. This third layer asks what can be learned about the physicochemistry of ncAA’s if we choose to assume that life’s standard amino acid alphabet represents an outcome of natural selection for a good set of “building blocks” with which to construct proteins?

## 4. The “Standard Alphabet” Is Distinctly Non-Random in Simple Ways

Over the past decade we and others have worked to quantify unusual properties of the standard alphabet of 20 amino acids that had evolved to become a universal feature of molecular biology by the time of LUCA [32]. Although two further amino acids, selenocysteine and pyrrolysine, appear to be in the mid to late stages of entering the genetic code for some lineages (see [33] for discussion), it is the unifying, standard alphabet of 20 that has remained the focus of attention. The interpretation offered is that distinctly non-random features are consistent with an outcome of natural selection for a particularly good set of building blocks with which to construct proteins.

An early challenge was to find relevant chemical descriptors with which to measure the standard alphabet of 20 genetically encoded amino acids relative to plausible alternatives. Whereas numerous quantitative measures described the sidechains of the standard alphabet [34], sparse and inconsistent data extended to other options. Even the most authoritative reviews of amino acid etiology focused on a case-by-case discussion of structures found within the standard alphabet [35]. In the latter years of the 20th century, however, powerful progress made primarily by drug discovery research [36] delivered algorithms capable of predicting accurately fundamental descriptors for chemical structures the size and complexity of individual amino acids [37]. Questions about unusual properties of the set could therefore narrow to looking for clear, non-random properties of the standard amino acid alphabet relative to other sets.

Three fundamental physicochemical properties of size, charge and hydrophobicity have received the most attention to date in identifying how the standard amino acid alphabet appears most clearly unusual (Figure 4). For each property, specific descriptors manifest non-randomness in two, simple statistics: range and evenness (together, “coverage”, Figure 4A). Specifically, the standard amino acid alphabet appears more evenly distributed across a broader range of values than can reasonably be explained by chance under definitions of increasing sophistication for what constitutes a superset of plausible alternative amino acid structures (Figure 4B–D).

An early study [41], for example, defined a set of 56 alternative amino acid structures to consider alongside the 20 of the standard alphabet: 42 structures detected within meteorites (an indication of prebiotic availability to life’s origins) and 14 more highly conserved biosynthetic intermediates (an indication of availability to molecular evolution, subsequent to life’s emergence). Calculating the range and evenness of 20-membered sets for a sample of 1 million alphabets drawn at random from this possibility space (_76_C_20_ ≅ 7.9 × 10^14^ sets of size 20) indicates a chance of 0.7% (±0.04) that 20 amino acids chosen at random would exhibit a larger range of values that are also more evenly distributed within this range for a given descriptor (Figure 4B). Given the strength of the signal but also the small set of amino acid structures used for comparison, subsequent work defined more carefully a comprehensive set of L-α-amino acid chemical structures worth considering as viable alternatives [42]. Retesting against this background revealed that among 10 million alphabets of size 20 drawn at random from a pool of ~1900 plausible chemical alternatives, only 6 alphabets exhibited a larger range and more even distribution in all three physicochemical properties (Figure 4C). In other words, this model indicates a probability of approximately one in two million that an amino acid set would exhibit better coverage by chance [39]. These results stimulated in turn a reinvestigation of the basis for thinking of size, charge and hydrophobicity as being relevant (including meta-analysis of template-free methods mentioned above which simplify amino acid alphabets to define physicochemical similarity) [43], along with efforts to find other quantitative predictions and tests for the underlying model [44] and discovery of further descriptors which yield fundamentally new insights [45].

The latest step in this line of research (Figure 4D) investigated variations in the choice of descriptors used to represent size, charge and hydrophobicity and also how range and evenness combine to create unusual coverage [40]. First, it seems that support for exceptional coverage is robust for hydrophobicity and volume (e.g., 1 × 10^−5^ = 0.001% chance of a random alphabet displaying better coverage in volume). For charge (pKa), unusual coverage does not apply to structures in which the amino and carboxyl termini of each amino acid have been capped so as to focus the descriptor value on the sidechain. This finding for pKa clearly challenges previous interpretations of selection for anything to do with protein folding. Second, it becomes clear that where exceptional coverage occurs, the effect is due primarily to EITHER exceptional range (for volume) OR exceptional evenness (for hydrophobicity)—though interaction effects are real and subtle. In fact, it seems that the range of hydrophobicities is actively constrained (not maximized). It is not yet known whether Figure 4D truly represents a simple, 6-dimensional model of what makes a good amino acid alphabet (perhaps 4-dimensional if pKa is truly a false trail), or whether selecting for any subset of these non-random attributes accounts for the others. In short, further questions abound, and that is the point, because they are tractable questions (Figure 5).

Before discussing which questions might be most useful to understanding ncAA’s, it helps to frame this lineage of research as an example of the Optimality Approach of evolutionary biology which did so much to advance an understanding of animal behavior during the late twentieth century by building quantitative models of adaptation [46]:

*“first we ask a question about why nature is doing something [Next we define] what we consider it possible for evolution to achieve … typically expressed as some constraints…an assumption must be made about what is being maximized … [this] optimization criterion is often an indirect measure of fitness [and] usually depends on trade-offs between … costs and benefits …The final step in the optimality approach is to test the predictions… If [the data] fit, then the model may really reflect the forces that have moulded the adaptation. If they do not, we may have misidentified the strategy set, or the optimization criterion, or the payoffs; or the phenomenon we have chosen may not in fact any longer be adaptive. **By reworking our assumptions, we modify our model and revise and retest the predictions**.”* [Emphasis added]

The authors explained their motivation to write this review of the optimality approach was to address the criticism that it is “*an iterative procedure leading inevitably to a fit*”. Their answer includes the point that a fit is only inevitable if one includes the possibility of growing constraints that reduce the perceived role for natural selection. Expressed this way, an increasingly good fit to observations of the real world formed by iteratively refining the assumptions of an explicit, quantitative model through predictions is what we often call scientific progress. The power of the optimality approach is that each iteration tends to suggest the next. Viewed in these terms, the series of publications investigating unusual properties of the standard alphabet spiral towards an increasingly refined quantitative model with which to understand what makes a good amino acid alphabet (Figure 5). Certainly, each publication establishes something worthwhile and new, but each brings to light limitations or flaws in the current model that lead to a new, improved version of the question.

Thus, for example, the latest step shown in Figure 4D may be used to seek out examples of the exceedingly rare combinations of 20 amino acids which exhibit better coverage (equal or larger range, and equal or lower evenness) than the standard alphabet. One such example is shown in Figure 6. This xenoalphabet exhibits superior coverage under the current definition of the term, but even a quick inspection of the structures involved shows that they tend to be larger and more hydrophobic than is common within the standard alphabet (Figure 6A). It is possible that the xenoalphabet shown might be capable of forming beta strands, but far less clear whether it could form alpha helices, or beta turns (although it does contain Cys and Ala). A Ramachandran plot for this alphabet would be new, relevant and exciting—although of course it would need to be generated computationally for the time being, perhaps by software such as that of Singh et al. [2], and perhaps in a framework like that presented by Kalmankar [15].

From an evolutionary perspective, future iterations of the model shown in Figure 5 could usefully incorporate into the fitness metric a penalty for total volumes exceeding that of the natural alphabet, and test whether perceptions of extraordinary coverage shift as a result. Here, or at any point, a better framing of the question could shift or eliminate what had previously seemed like good design by natural selection. Again, asking “what seems to have been optimized?” does not inevitably find natural selection, because it could instead help to reveal constraints: done carefully, the only inevitability is to make progress in understanding the phenomenon under scrutiny. However, this particular evolutionary question (of small volume amino acids) may be less relevant to synthetic biology because plotting ncAA’s alongside the computational library (Figure 7) clarifies that, so far, most of the ncAA’s which have been incorporated experimentally into ribosomal peptide synthesis are far larger than anything within the computational library. The same plot also reveals that ncAA’s are generally biased towards hydrophobic sidechains (possibly as a result of often favoring aromatic structures): a clear region of large, hydrophilic sidechains in the computational library is currently unexplored by empirical ncAA’s.

Potential “better” alphabets located in any of the regions populated in Figure 7 (current, bulky alphabets; bulkier alphabets made from known ncAA’s; and bulky alphabets which are less hydrophobic) are all relevant to the world of ncAA protein structure. Even underpopulated regions of amino acid chemical space shown in Figure 7 deserve careful exploration for plausible structures that have been overlooked by current, computational methodology [42]. Exploration and analysis are warranted if only because detecting why a theoretical, better alphabet does not function well for protein synthesis informs a better definition of what makes a successful set of building blocks. Ultimately these alphabets, and the questions they provoke, must become much better understood through experimental work that tests the hypotheses generated by theory. There are, however, many further computational and theoretical tools that could be brought to bear before and during any such empirical work, including (but not limited to) those which further refine of the model of what natural selection was up to in the days before LUCA.

## 5. Conclusions

This review summarizes and juxtaposes two different research frontiers that have emerged from two different academic cultures and perspectives. One is the largely empirical world of ncAA incorporation via synthetic biology; the other is a largely theoretical world of exploring amino acid alphabet etiology. It is not a novel observation that questions about life’s origins share much in common with questions of synthetic biology (see, for example, [47]) but our intention here is to stimulate a new, direct and specific interface between experts from these two worlds in order to progress science regarding the design of amino acid alphabets.

## Figures and Tables

**Figure 1 ijms-22-02787-f001:**
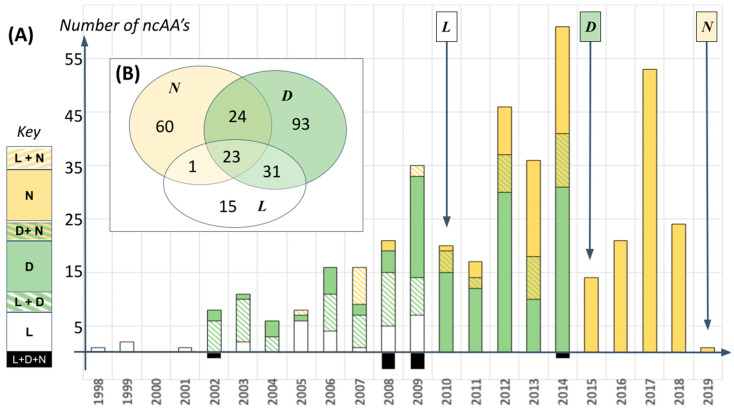
Number of non-canonical amino acid (ncAA) structures reported in peer reviewed publications, according to three major reviews of the topic which describe, collectively, 251 papers or patents and 246 unique structures: L = Liu & Schultz (2010) [16]; D = Dumas et al. (2015) [17]; N = Nodling et al. (2019) [18]. (**A**) Publications by year: different colors distinguish the number and overlap of ncAA structures reported by each review. (**B**) Venn diagram of total overlap/unique structures between each review.

**Figure 2 ijms-22-02787-f002:**
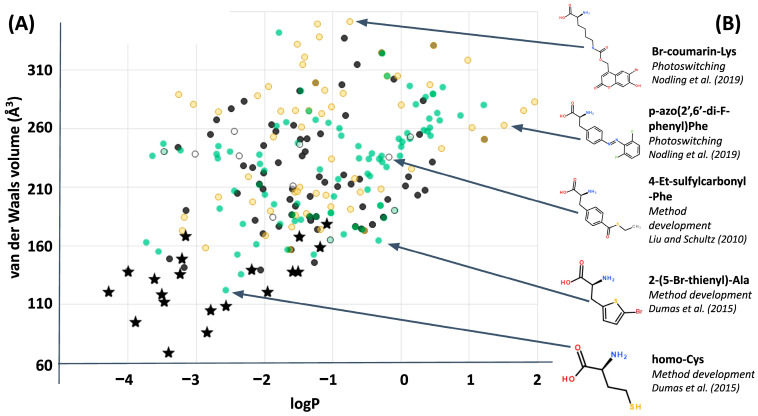
Empirical additions to the alphabet of genetically encoded amino acids. (**A**) a plot of size (calculated as van der Waals volume) and hydrophobicity (LogP) shows that ncAA’s (colored circles) are usually larger and more hydrophobic than the members of the standard amino acid alphabet (black stars). (**B**) Five examples of ncAA structures.

**Figure 3 ijms-22-02787-f003:**
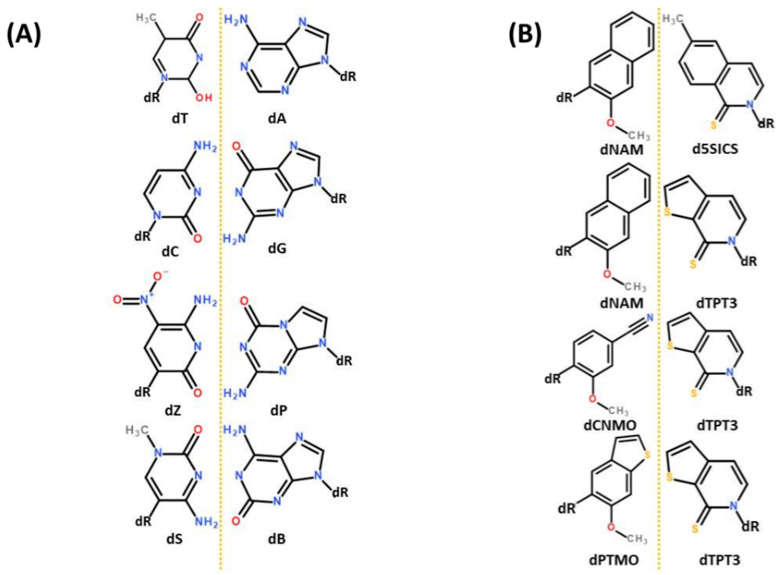
Two very different synthetic nucleotide alphabets developed to the point of in vivo functioning. (**A**) The *Hachimoji* alphabet [20] emulates Watson-Crick base pairing using the four atom types known to “natural” nucleotides (N, O, H and C). (**B**) Romesberg and colleagues’ genetic alphabet [21] achieves base pairing through hydrophobic and packing forces and uses sulfur, an atom type unknown to ‘natural’ genetics.

**Figure 4 ijms-22-02787-f004:**
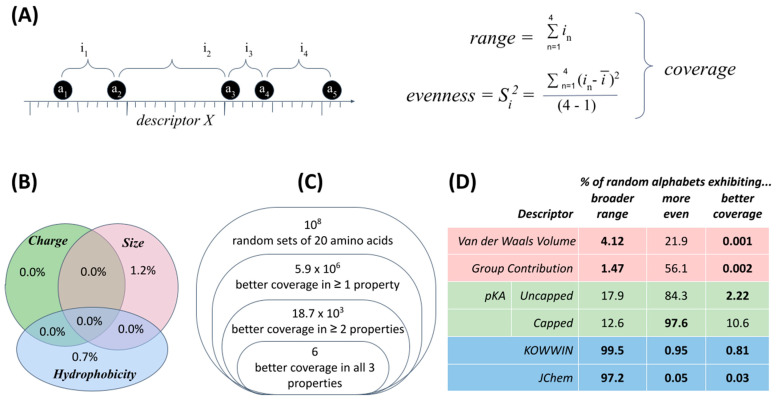
Unusual properties of the standard amino acid alphabet. (**A**) For a given chemical descriptor such as van der Waals volume, “coverage” combines two statistics to represent a set of amino acids. (**A**) Illustrates these statistics for a set of five amino acids (a_1_…a_5_) with four corresponding intervals (i_1_…i_4_) measured in terms of the hypothetical quantitative descriptor *x*. Evenness is the sample variance (S^2^) of intervals between neighboring amino acids; “Range” is the sum of these intervals (∑i_1…4_); (**B**) Percentage of random amino acid alphabets drawn from a pool of 76 plausible alternatives that are more evenly distributed over a larger range than the standard alphabet, data from [38]; (**C**) Number of randomized amino acid alphabets within a sample of 10 million which are more evenly distributed over a broader range of values for one, two and all three of these properties, data from [39]; (**D**) the relative contribution of range versus evenness for variations in each descriptor, adapted from [40], with all significant values (<5%) shown in bold font.

**Figure 5 ijms-22-02787-f005:**
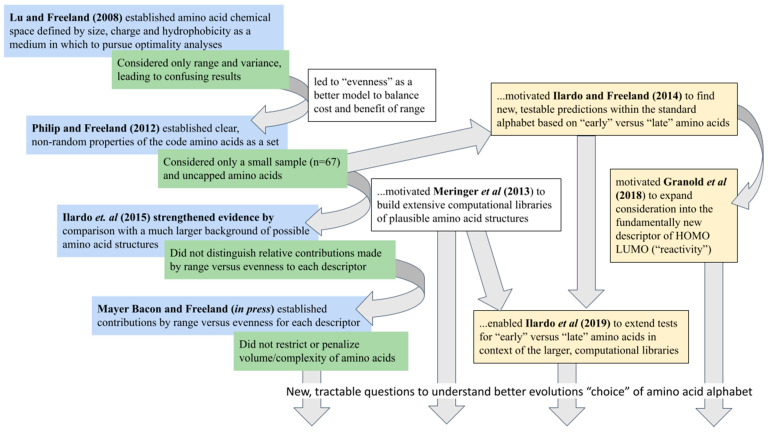
The series of publications reporting unusual features of physicochemistry for the standard amino acid alphabet may be viewed usefully as a series of flawed models attempting to answer the question “What exactly seems to have been optimized?” Following the optimality approach of evolutionary biology [46], the model grows more complex at each step, but only in specific, targeted attempts to address these flaws as they come to light. Each step reveals weaknesses in the assumptions thus far, stimulating subsequent steps in a process that refines the model by introducing statistical complexity only as needed.

**Figure 6 ijms-22-02787-f006:**
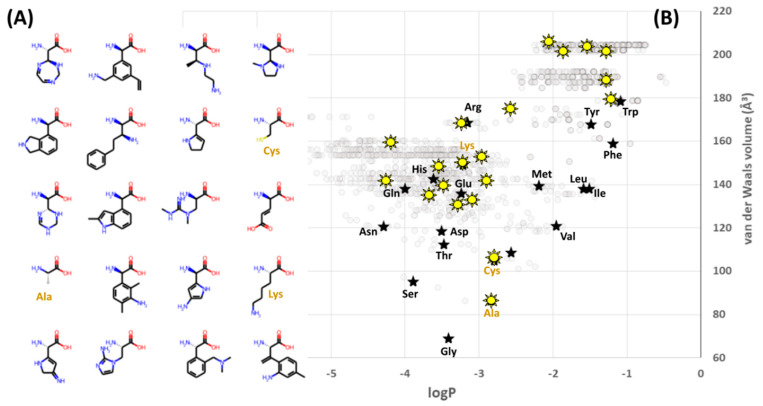
A “better” amino acid alphabets and its context: (**A**) an example of chemical structures for one of the rare sets of 20 amino acids that exhibits better coverage (broader range and more evenness within this range) for size and hydrophobicity; (**B**) the context of amino acid structures within which this alphabet occurs: members of the better alphabet (yellow stars) are shown alongside the members of the standard alphabet (black stars, labelled with 3-letter abbreviations) and the library of plausible structures (gray circles) within which this better alphabet was found.

**Figure 7 ijms-22-02787-f007:**
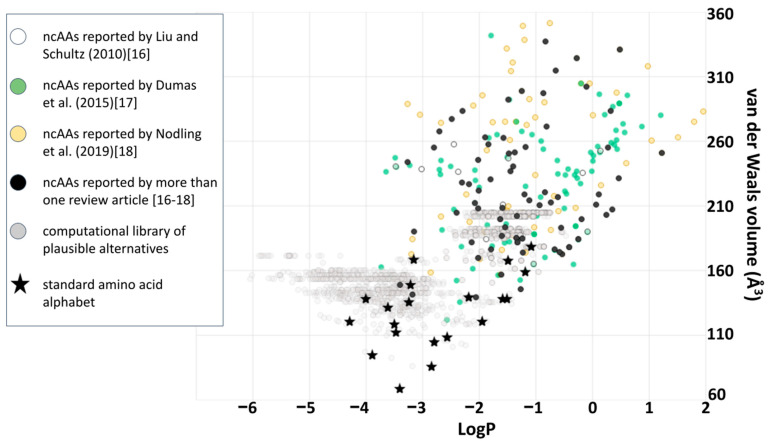
Genetically encoded amino acids as a small subset of chemically plausible alternatives. A comparison of ncAA’s (colored circles, sources and data as per Figure 2) with the computational library of 1913 amino acids used to derive results shown in Figure 4C,D and the 20 amino acids of the standard amino acid alphabet (see also Figure 6).

## Data Availability

Two supplemental data files are available as Appendix A (above).

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
