# Peer review of "Evolution as a Guide to Designing *xeno* Amino Acid Alphabets"

_ijms, 2021, doi:10.3390/ijms22062787_

Round 1
Reviewer 1 Report
This theoretical review addresses the question of whether the 22 amino acids "chosen" by natural selection is the best one to give rise to a the great diversity of functional proteins, or another "amino acid alphabet" could be as good to reach this aim.
The authors could develop this particular question in more details, taken into account the theoretical models they describe.
Minor comments :
- ncAA abbreviation : decide between "non-canonical amino acids" and "non-coded amino acids".
- line 210 : actually the standard alphabet contains 22 amino acids. In some circumstances the nonsense (or stop) codons may code for an amino acid (UGA for selenocysteine; UAG for pyrrolysine).
- Figure 6 : Indicate (A) and (B) in the figure.
Reviewer 2 Report
The work by Mayer-Bacon et. al. is an important perspective on the design of xenoprotein systems. Although the work is scientifically sound, there are minor factors that should be described in a more in-depth manner.
Major points:
- The paragraph stating: “When it comes to proteins comprising ncAA’s, however,...” where authors suggest that solutions based on the machine learning are limited to their learning domain and thus cannot extrapolate to properly work with ncAA’s (“It is unsafe to assume that machine learning trained on a given library of proteins can extend predictions reliably to sequences far removed from anything in that library"). This statement is true as machine learning models had severe issues with extrapolating outside of their learning domain. However, some machine models are not based directly on amino acids, but rather on their physicochemical representations either as numerical values or simplified alphabets where similar amino acids are grouped together based on their properties (as here 10.1093/bioinformatics/btp164, 10.1016/j.jtbi.2015.07.024 or 10.1038/s41598-017-13210-9). This reviewer recommends adding a new paragraph discussing the possibility of incorporating ncAA’s into machine learning models based on the simplified amino acid alphabets.
- This reviewer considers the author's statement on the template-free modeling “The greater problem for template free prediction is the number of possible conformations that require comparison.” as true but oversimplified. Despite their limitation, template-free modeling sometimes outperforms template-based solutions (10.1093/bib/bbt047). Given the relatively good performance of template-free prediction, the additional layer of complication in the form of ncAA’s poses an addressable challenge.
- The article skims over the topic of xenoprotein engineering. This reviewer suggests mentioning challenges in experimental studies (like 10.1073/pnas.1722633115) aside from the purely computational solutions.
Minor points:
- The paragraph “The limitations of template-based...” ends with an atypical citation “Zagrovic review”.
